# Effect of Trimethine Cyanine Dye- and Folate-Conjugation on the In Vitro Biological Activity of Proapoptotic Peptides

**DOI:** 10.3390/biom12050725

**Published:** 2022-05-20

**Authors:** Davide Cardella, Wenjing Deng, Louis Y. P. Luk, Yu-Hsuan Tsai

**Affiliations:** 1School of Chemistry, Main Building, Cardiff University, Park Place, Cardiff CF10 3AT, UK; cardellad@cardiff.ac.uk; 2Shenzhen Bay Laboratory, Gaoke Innovation Center, Institute of Molecular Physiology, Guangming District, Shenzhen 518132, China; dengwenjing111@i.smu.edu.cn

**Keywords:** cancer, cyanine dyes, folate, mitochondria, proapoptotic peptides, solid-phase synthesis

## Abstract

Despite continuous advances, anticancer therapy still faces several technical hurdles, such as selectivity on cellular and subcellular targets of therapeutics. Toward addressing these limitations, we have combined the use of proapoptotic peptides, trimethine cyanine dye, and folate to target the mitochondria of tumor cells. A series of proapoptotic peptides and their conjugates with a cyanine dye and/or folate were synthesized in the solid phase, and their toxicity in different human cell lines was assessed. Cyanine-bearing conjugates were found to be up to 100-fold more cytotoxic than the parent peptides and to localize in mitochondria. However, the addition of a folate motif did not enhance the potency or selectivity of the resulting conjugates toward tumor cells that overexpress folate receptor α. Furthermore, while dual-labeled constructs were also found to localize within the target organelle, they were not generally selective towards folate receptor α-positive cell lines in vitro.

## 1. Introduction

Cancer is a leading cause of death worldwide [1]. It is estimated that more than 50% of the people in the UK who are currently under the age of 65 will be diagnosed with cancer during their lifetime [2]. While significant research has been conducted in therapeutic development, the treatment of different cancers still remains an unmet clinical need [3,4]. Thus, there is a necessity for new therapeutic approaches, particularly ones that specifically target cancer cells [5,6]. Such a strategy can maximize potency and minimize off-target side effects of the drugs.

The mitochondrion hosts several putative drug targets of cancer therapy, as the organelle plays key functions in different physiological and pathological cellular processes such as programmed cell death, and hence small-molecule agents that target this organelle are of interest [7]. Many heptamethine cyanine dyes are preferentially localized in the mitochondria of cancer cells and have been employed as imaging probes [8,9,10,11]. The mitochondrial targeting ability is believed to be associated with the delocalized lipophilic cationic nature of the dye as depolarization of the mitochondrial membrane prevents accumulation of heptamethine cyanine dyes in mitochondria [12], while selectivity towards cancer cells is thought to be due to the generic negative potential of the cell surface [13]. Moreover, we recently showed that Cy3, a trimethine cyanine dye, can function as a delivery vector selectively targeting mitochondria with a moderate preference toward tumor cells, and its affinity also depends on the mitochondrial membrane potential [14]. It, therefore, is of interest to investigate whether the addition of a cell-targeting component could further improve the cell selectivity of the conjugates.

Folate receptor α, because of its role in uptaking the essential vitamin, is overexpressed in many tumors and hence can be targeted for cargo delivery. To date, many anti-cancer therapeutics have been preferentially delivered to tumor cells upon conjugation to folate [15,16,17,18,19]. Some of the most successful examples reported in the literature are the conjugation of folate to desacetyl vinblastine monohydrazine [20] and taxol derivatives [21]. In both cases, the drug and folate component were separated by a peptidyl spacer and a self-cleaving linker. Additionally, folate conjugation was also proven to be effective in the delivery of antibodies, nanoparticles, and imaging agents [16,17,18].

Here we tested the hypothesis of whether the covalent addition of folate and cyanine dye will enhance the potency and selectivity of toxic cargoes towards mammalian cells, including cancer cell lines (Figure 1a). Cyanine dye **1** (Figure 1b) and folate were chemically conjugated to the N-terminus of three proapoptotic peptides (**2**–**4**) to yield conjugates **5**–**13** (Table 1). The cytotoxicity of these constructs was evaluated in cancer (i.e., KB, MCF7, and SK-OV-3) and non-cancer (i.e., HEK293) cell lines.

## 2. Materials and Methods

Synthesis of compounds **1**–**13** is described in the Appendix A.

### 2.1. Cell Culture

KB and MCF7 cells were kindly gifted by Prof. Arwyn T. Jones (Cardiff University), HEK293 cells were purchased from Public Health England, and SK-OV-3 cells were purchased from National Collection of Authenticated Cell Cultures (https://cell-bank.org.cn/). Cells were routinely tested for mycoplasma infection. KB and SK-OV-3 cells were maintained in T75 flasks at 37 °C in a 5% CO_2_ atmosphere in folic acid-depleted Roswell Park Memorial Institute (RPMI)-1640 medium supplemented with 10% (*v*/*v*) fetal bovine serum (FBS). MCF7 and HEK293 cells were maintained in T75 flasks at 37 °C in a 5% CO_2_ atmosphere in Dulbecco’s Modified Eagle’s medium (DMEM) supplemented with 10% (*v*/*v*) FBS. Cells were maintained at a sub-confluent monolayer and split at 80–85% confluency. For splitting, cells were washed with PBS, trypsinized in 1 mL of trypsin, and 200 µL of the 1000 µL trypsin cell suspension was re-suspended in 12 mL fresh DMEM containing 10% (*v*/*v*) FBS in a new T75 flask. KB and SK-OV-3 cells were used for biological experiments after being maintained for at least 10 days in folic acid-depleted RPMI-1640 medium.

### 2.2. Cell Viability Assay

SK-OV-3, MCF7, and HEK293 cells were seeded at a density of 2 × 10^4^ cells per well in a Nunc™ MicroWell™ 96-well plate and grown at 37 °C in a 5% CO_2_ atmosphere in DMEM supplemented with 10% (*v*/*v*) FBS for 24 h. KB cells were seeded at a density of 7 × 10^3^ cells per well in a 96-well plate and grown at 37 °C in a 5% CO_2_ atmosphere in folic acid-depleted RPMI-1640 medium supplemented with 10% (*v*/*v*) FBS for 24 h. Stock solutions of compounds **2**–⁠**6** and **11**–**13** were obtained by dissolving the compounds in sterile deionized water. Compounds **1**, and **7**–**10** were dissolved in pure DMSO. The stock solutions were diluted into the proper medium (according to the cell line tested) supplemented with 10% FBS to the appropriate concentration, and cells in each well were incubated with 100 µL of the solution. The solutions in each well were then adjusted to a concentration of 1% (*v*/*v*) DMSO. After 24 h at 37 °C, 20 µL of CellTiter-Blue™ was added to each well. The plate was incubated for another 4 h at 37 °C before analysis on a Perkin Elmer Victor X plate reader (excitation 531 nm; emission 595 nm). Each data point was calculated from a minimum of nine values resulting from three biological replicates (i.e., cells split from three different passages); each biological replicate is calculated from three technical replicates (i.e., cells split from the same passage). Value from media-only with CellTiter-Blue™ was set at 0% viability. This value was then subtracted from the values from cell-only (i.e., non-treated) wells with CellTiter-Blue™ in each biological replicate and set as 100% viability. For treatments of Cy3-containing compounds, blanks were generated with cell-free wells containing the compounds incubated for 24 h before addition of CellTiter-Blue™. The fluorescent reading for these wells was deducted from the treatment readings. Dose-response fitting curves were generated using Origin, version 2019b, OriginLab Corporation, Northampton, MA, USA (Appendix A).

### 2.3. Confocal Microscopy

SK-OV-3 and MCF7 were seeded at a density of 1 × 10^6^ cells per well in glass-bottom culture dishes and grown at 37 °C in a 5% CO_2_ atmosphere in medium supplemented with 10% (*v*/*v*) FBS for 24 h. Cells were washed once with the RPMI-1640 medium supplemented with 10% (*v*/*v*) FBS before staining. Compounds **1**, **5**–**10** were diluted into the RPMI-1640 medium supplemented with 10% (*v*/*v*) FBS to 10 μM and then added to the culture dishes, respectively. After 10-min incubation at 37 °C, cells were washed with RPMI-1640 medium supplemented with 10% (*v*/*v*) FBS. Cells were then treated with 50 nM MitoTracker Green FM and 10 μg/mL Hoechst33258 pre-formulated with RPMI-1640 medium supplemented with 10% (*v*/*v*) FBS and then incubated for another 10 min at 37 °C before analysis on ZEISS LSM 900 with Airyscan 2. The results were analyzed using Image J software (Figure 3 and Appendix A).

### 2.4. Flow Cytometry

SK-OV-3 and MCF7 were seeded at a density of 2 × 10^5^ cells per well in 24-well plate and grown at 37 °C in a 5% CO_2_ atmosphere in DMEM medium supplemented with 10% (*v*/*v*) FBS for 24 h. Cells were washed once with the RPMI-1640 medium supplemented with 10% FBS before staining. Compounds **1**, **5**–**10** were diluted into the RPMI-1640 medium supplemented with 10% FBS to 10 μM and then added to the culture dishes. After 10-min incubation at 37 °C, cells were washed with RPMI-1640 medium supplemented with 10% FBS. Cells were then treated with 50 nM Mitotracker Green FM and 10 μg/mL Hoechst33258 pre-formulated with RPMI-1640 medium supplemented with 10% FBS for another 10 min at 37 °C. Cells were washed once with PBS before trypsinization for flow cytometry analysis on Invitrogen Attune NxT. The results were analyzed using FlowJo™ v10 Software of BD Life Sciences (Appendix A).

### 2.5. Statistical Analysis

Comparison of cytotoxicity fitting curves was achieved by applying the extra sum-of-squares F test, performed using GraphPad Prism version 9 for Windows, GraphPad Software, La Jolla, California, USA (https://www.graphpad.com/). Statistical difference between cytotoxicity values was assumed when *p*-value, as output of the extra sum-of-squares F test, was found to be below 0.05. Statistical analysis on the flow cytometry data was achieved by applying the unpaired two-tailed *t*-test test, performed using GraphPad Prism version 9 for Windows, GraphPad Software, La Jolla California USA, USA (https://www.graphpad.com/).

## 3. Results

### 3.1. Synthesis of Peptides and Peptide Conjugates **2**–**13**

Peptides **2**–**4** were synthesized on a polystyrene-based Rink amide resin using an automated microwave peptide synthesizer. Initial attempts using 2.25 min/coupling yielded both the desired peptide and a truncated side product that lacks an N-terminal lysine residue (Appendix A). This issue was addressed by extending the coupling time (see Appendix A).

The peptidyl backbone of conjugates **5**–**7** was synthesized following the optimized protocol for the synthesis of peptides **2**–**4**. After coupling and deprotection of the last amino acid residue, the peptidyl resin was manually coupled to cyanine dye **1**.

For the synthesis of folate-containing conjugates **8**–**13**, an extra lysine was added at the N-terminal of the peptidyl backbone, and cyanine dye **1** and folate were coupled to its α- and ε-amine groups, respectively. A literature procedure was first attempted [22]. However, this strategy led to folate conjugation at both its α- and γ-carboxylate groups, giving an inseparable mixture of labeled isomers (Appendix A). To overcome this issue, a protected glutamic acid residue, Fmoc-Glu(OtBu)-OH, was first coupled to the peptide, followed by coupling with pteroic acid. Since no difference was found in the endocytosis efficiency of conjugates where the folate is labeled through either its α- or γ-carboxylic group [23], we opted for the reaction of the α-carboxylate due to the ready availability of the reagent. As a protecting group for the extra lysine side chain to be conjugated with folate, 4-methyltrityl (Mtt) was chosen for conjugates **8**, **9**, and **11**–**13**. In this way, the protecting group on the N-terminal lysine could be selectively cleaved with conditions orthogonal to the other protecting groups on the peptidyl resin. However, the synthesis of compound **10** proved to be more challenging. In fact, upon glutamic acid coupling and Fmoc deprotection, the Mtt strategy led to the presence of a peak in the LC-MS chromatogram with a difference of + 129 m/z compared to the desired intermediate, attributable to an extra glutamic acid residue (Appendix A). This is likely due to Boc deprotection during the Mtt cleavage conditions (1% TFA in DCM). Finally, conjugate **10** was obtained using 1-(4,4-dimethyl-2,6-dioxocyclohex-1-ylidene)-3-methylbutyl (ivDde) as a protecting group, which could be cleaved in 4% (*v*/*v*) hydrazine hydrate in DMF, leading to selective labeling of one folate molecule to the peptide.

### 3.2. Cytotoxicity of **1**–**13**

Cytotoxicity of **1**–**13** was evaluated using the CellTiter-Blue assay on KB, MCF7, SK-OV-3, and HEK293 cells. KB cells are derived from human cervical cancer [24], while SK-OV-3 cells are derived from human ovarian cancer. Both cell lines exhibit high levels of folate receptor α on the cell surface [17,25,26]. This feature is of particular relevance in anticancer applications, as it can be exploited for selective targeting and delivery. MCF7 is a cell line derived from human breast cancer with low levels of folate receptor α [27]. Therefore, it can serve as the negative control for folate receptor α mediated tumor-targeted drug delivery systems. HEK293 is a non-cancer cell line and was chosen to evaluate the specificity of the tested compounds toward cancer cells. Cell viability assay results are summarized in Table 1.

Peptides **2** and **3** were found to be the least potent amongst the tested compounds, whereas peptide **4** showed good potency in all tested cell lines. In fact, peptide **4** was designed by engineering the sequence of compounds **2** and **3** via the replacement of the leucine with cyclohexylalanine residues which led to increased cytotoxicity [28]. Cyanine dye-labeled constructs **5**–**7** were found to be significantly more potent (*p* < 0.01) than the peptides alone (**2**–**4**) in the tested cell lines—the only exception being the cytotoxicity of compound **7** in KB cells which was not found to be significantly different (*p* = 0.11) to that of its native sequence. Conjugates **11** and **12** bearing a folate component showed enhanced toxicity in KB but not in SK-OV-3 cells when compared to MCF7 and HEK293 cells. Likewise, the cytotoxicity of compound **13** in KB cells was found to be significantly higher (*p* < 0.01) than in the other tested cell lines. Nevertheless, the potency of **11**–**12** in cells not overexpressing folate receptor α is comparable to that of their native peptides. Lastly, dual labeled conjugates **8**–**10** showed enhanced potency compared to the parent sequences and to compounds **11**–**13** (*p* < 0.05), although no selectivity was consistently observed toward KB and SK-OV-3 cells (Figure 2).

### 3.3. Mitochondrial Localization and Cellular Uptake of **5**–**10**

To confirm the subcellular localization and uptake of Cy3-containing conjugates, we performed confocal microscopy and flow cytometry analysis of cyanine dye **1** as well as conjugates **5**–**10** in SK-OV-3 and MCF7 cells. Confocal microscopy results confirmed the mitochondrial localization of **5**–**10** (Figure 3 and Appendix A in Appendix A). Flow cytometry analysis showed similar uptake levels of individual conjugates by the two different cell lines (Figure 4 and Appendix A), but conjugates **8**–**10** containing both Cy3 and folate were less efficient in entering cells than conjugates **5**–**7** containing only Cy3. Overall, the less cytotoxic conjugates also have a lower level of cellular uptake. 

## 4. Discussion and Conclusions

Both organelle-specific and receptor-mediated drug delivery systems have proven to be promising tools in anticancer therapy. We have previously shown the efficacy of a trimethine cyanine dye in selectively delivering different cargos to the mitochondria of human cancer cell lines [14]. Many cancer cells overexpress the folate receptor α [17], enabling the use of folate to target cancer cells, especially those derived from ovarian, breast, and lung carcinomas [16]. With these premises, we sought to improve the efficacy of existing proapoptotic peptides **2**–**4** toward cancer cells via conjugation with both cyanine dye and folate (**8**–**10**). For systematic evaluation, conjugates containing either cyanine dye (**5**–**7**) or folate (**11**–**13**) were also prepared.

The syntheses were accomplished with as little as two equivalents of Fmoc-protected amino acid per coupling, whereas literature procedures for preparing peptides **2** and **3** employed four or more equivalents [22,29]. This is of great relevance when expensive d-amino acid or unnatural amino acid (e.g., cyclohexylalanine) building blocks are used.

Cytotoxicity results obtained by the CellTiter-Blue assay are shown in Table 1. The low activity of peptides **2** and **3** in the tested cell lines (EC_50_ > 300 µM) is likely due to their low cell permeability [29,30,31,32,33,34,35]. In contrast, peptide **4** was engineered to be more hydrophobic than **2** or **3** by replacing leucine with cyclohexylalanine to improve membrane permeability [28]. Indeed, peptide **4** is significantly more toxic than **2** or **3**, and the obtained EC_50_ values align with the literature [28], confirming the correlation between hydrophobicity and cell permeability [28,36,37]. Interestingly, the EC_50_ value of **4** in MCF7 cells was about 3-8 times higher than in the other tested cell lines.

Delocalized lipophilic cations, such as the triphenylphosphonium group and the cyanine dyes used here, preferentially localize within mitochondria. While the triphenylphosphonium group is widely used to generate mitochondrial-targeting molecules, conjugation of peptide **2** with the triphenylphosphonium group showed negligible toxicity in mammalian cells [22]. In contrast, cyanine dye-labeled constructs **5**–**7** were generally more potent than the peptides alone (**2**–**4**) in the tested cell lines. The enhanced potency of these constructs may be due to the increased mitochondrial targeting ability and/or membrane permeability upon conjugation with cyanine dye **1**. Indeed, the confocal microscopy results confirmed the mitochondrial localization of the cyanine dye-labeled constructs (Figure 3).

We were intrigued by the enhanced potency caused by cyanine dye conjugation; we aimed to increase the selectivity of the apoptotic peptides towards cancer cells. Therefore, dual-labeled constructs **8**–**10** were made with the α-amine of the N-terminal lysine conjugated to cyanine dye **1** and the side chain of the same lysine connected to a folate. As a control, we also prepared and tested the activity of compounds **11**–**13**, where only folate but not the cyanine dye was included.

To our disappointment, folate-containing constructs (**8**–**13**) did not display the expected selectivity towards KB and SK-OV-3 cells, regardless of the presence of a cyanine dye motif. Conjugates **11**–**13** containing only folate but not cyanine dye have comparable potency to the parent sequences **2**–**4** towards MCF7, SK-OV-3, and HEK293 cells. Similarly, the addition of folate to Cy3-containing conjugates **5**–**7** has minimal impact on selectivity and potency. Overall, the attachment of a folate motif through its α-carboxylic group to the N-terminus of the peptides did not lead to any notable selectivity towards KB and SK-OV-3. Optimizing the attachment site, linkage, linker, and valency of folate is likely essential for receptor-mediated uptake, conferring tumor cell selectivity.

Previously, attachment of a folate motif to the N-terminus [22,38,39], C-terminus [40], or in the middle of peptides [41] has led to either enhanced selectivity of the construct towards cells overexpressing the folate receptor α or enhanced binding affinity to recombinant folate receptor α. It has been reported that linking the folate via the α- or γ-carboxylic group does not affect the uptake efficiency [23]. However, a different study claimed otherwise [42], and, in most of the successful examples, the folate is connected through its γ-carboxylic group [17]. Many of these constructs also bear a spacer between the folate and the cargo [17]. Although the effects of the linker length and type have not been systematically evaluated, the steric environment around the folate fragment is known to be an important factor in the interaction of the conjugate with the receptor [23]. Besides, some proteins have shown improved uptake by folate receptor α-positive cells when labeled with multiple folate molecules [43,44]. For future work, the attachment site, linkage, linker, and valency of the folate within the constructs will be evaluated.

In conclusion, cyanine dyes are promising tools to further improve the efficacy of drugs in anticancer therapy. Nevertheless, further effort must be made to optimize the design of constructs bearing both components.

## Figures and Tables

**Figure 1 biomolecules-12-00725-f001:**
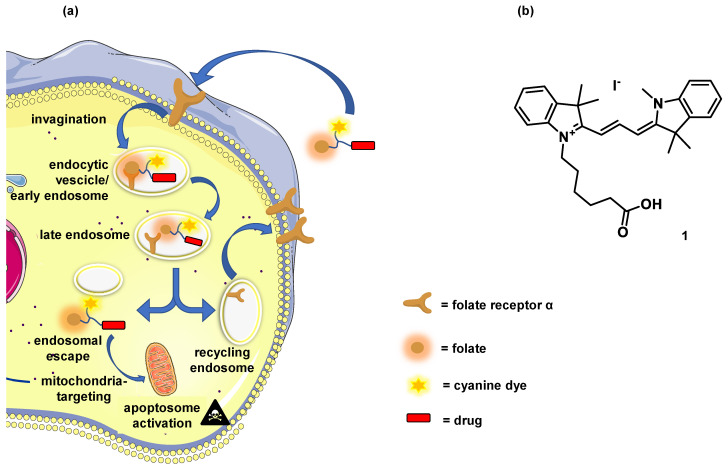
Folate conjugates exploit the folate receptor-mediated endosomal pathways, while the cyanine dye component delivers the construct inside mitochondria. (**a**) Folate receptor-mediated endocytosis mechanism and cyanine dye-mediated mitochondria-targeting in mammalian cells; (**b**) Chemical structure of cyanine dye **1**.

**Figure 2 biomolecules-12-00725-f002:**
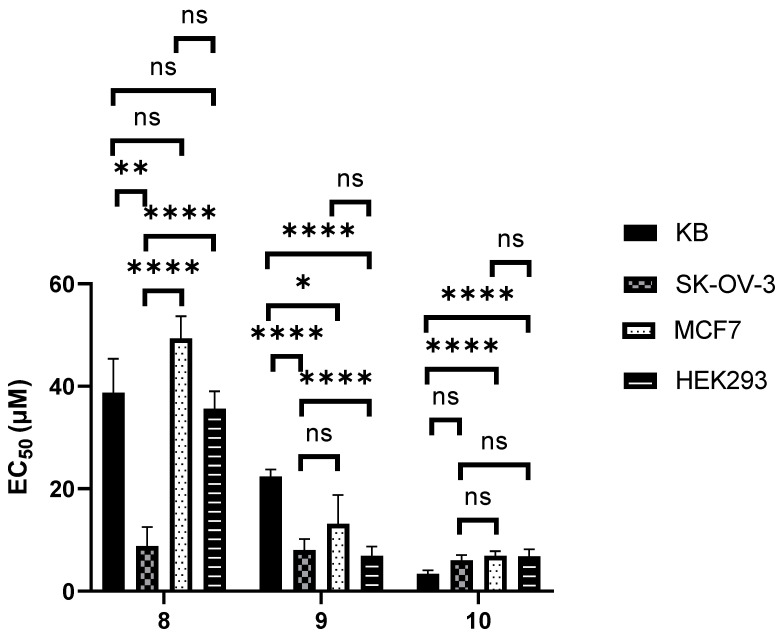
Cytotoxicity of cyanine dye and folate labelled compounds **8**–**10** in KB, SK-OV-3, MCF7 and HEK293 cells. Bars represent the standard error of the curve fitted using Origin. Statistical analysis was performed by applying the extra sum-of-squares F test, performed using GraphPad Prism version 9 for Windows, GraphPad Software, La Jolla California USA, www.graphpad.com. ns = not statistically significant, * = *p* < 0.05, ** = *p* < 0.01, **** = *p* < 0.0001.

**Figure 3 biomolecules-12-00725-f003:**
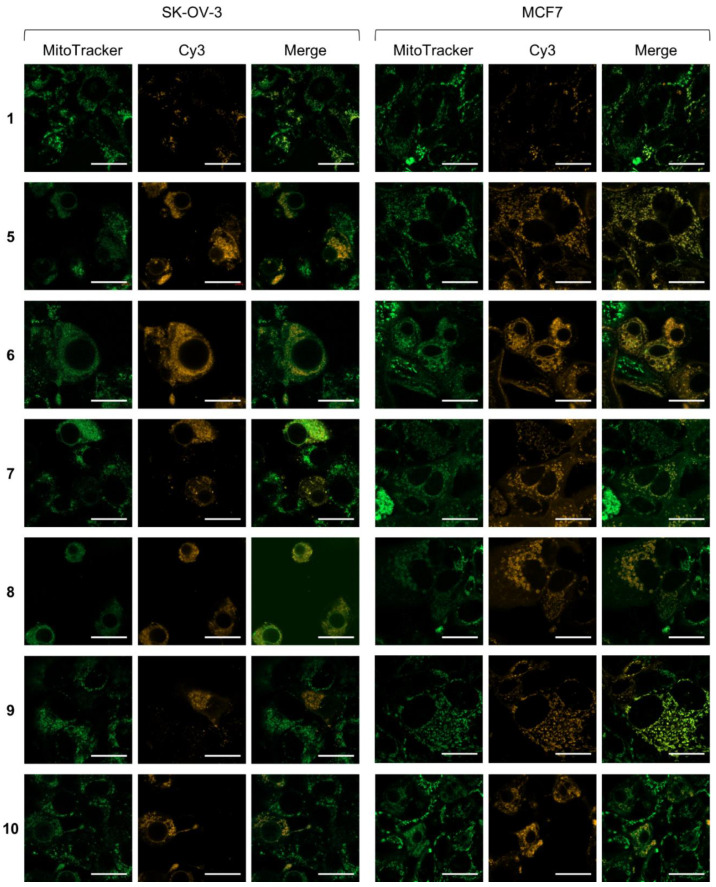
Confocal microscopy images of SK-OV-3 and MCF7 cells treated with Cy3-containing conjugates. Cells were treated with 10 µM of the indicated conjugate for 10 min at 37 °C, washed, stained with 50 nM MitoTracker Green FM and 10 μg/mL Hoechst33258 for 10 min at 37 °C, washed, and imaged at 63X. Pearson’s correlation coefficients of MitoTracker Green FM and Cy3 fluorescence for compounds **1**, **5**, **6**, **7**, **8**, **9**, and **10** in SK-OV-3 and MCF cells are 0.75/0.64, 0.73/0.81, 0.91/0.61, 0.78/0.63, 0.85/0.69, 0.43/0.82, and 0.78/0.60, respectively. Excitation wavelengths for Hoechst33258, MitoTracker Green FM and Cy3 were set as 405, 488, and 561 nm, respectively. Scale bars denote 25 μm. Additional images, including the results of Hoechst33258 staining, are shown in Appendix A (SK-OV-3 at 20X), Appendix A (SK-OV-3 at 63X), Appendix A (MCF7 at 20X) and Appendix A (MCF7 at 63X), Appendix A.

**Figure 4 biomolecules-12-00725-f004:**
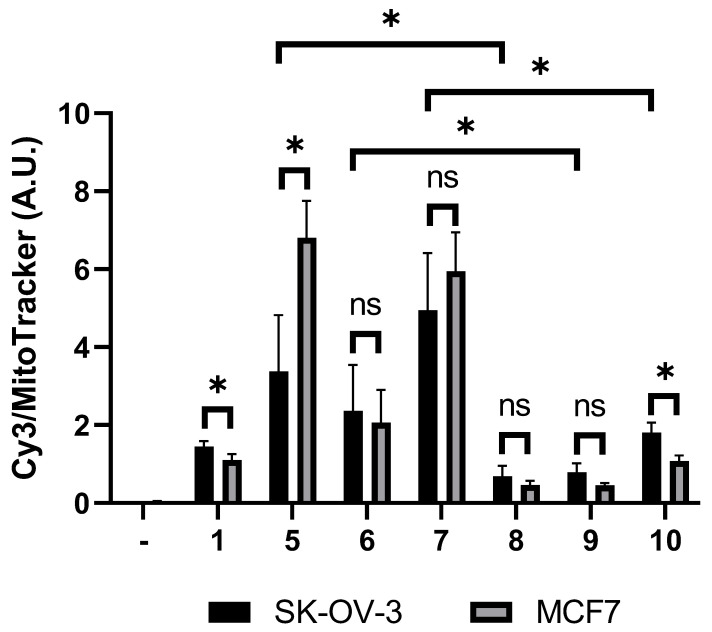
Flow cytometry analysis of Cy3-containing conjugate uptakes by SK-OV-3 and MCF7 cells. Cells were treated with 10 µM of the indicated conjugate for 10 min at 37 °C, washed, stained with 50 nM MitoTracker Green FMfor 10 min at 37 °C, washed, trypsinized, and subjected to flow cytometry analysis. Cellular uptakes are shown as the ratios of Cy3 and MitoTracker Green FM fluorescence. Mean ratios ± standard deviations are calculated from three biological replicates and plotted in arbitrary unit (A.U.). The negative control (-) refers to cells only stained with MitoTracker Green FM. Representative flow cytometry scatter plots are shown in Appendix A. Statistical analysis was performed by applying the unpaired two-tailed *t*-test test, performed using GraphPad Prism version 9 for Windows, GraphPad Software, La Jolla California USA, www.graphpad.com. ns = not statistically significant, * = *p* < 0.05.

**Table 1 biomolecules-12-00725-t001:** Chemical structure of **2**–**13** and cytotoxicity values for compounds **1**–**13**. The EC_50_ values (expressed in µM) on KB, SK-OV-3, MCF7 and HEK293 cells of different molecules and their cyanine dye conjugates were quantified using cell viability assays. Data from three biological replicates were combined and fitted using Origin to obtain the EC_50_ values and the standard errors (shown in the brackets).

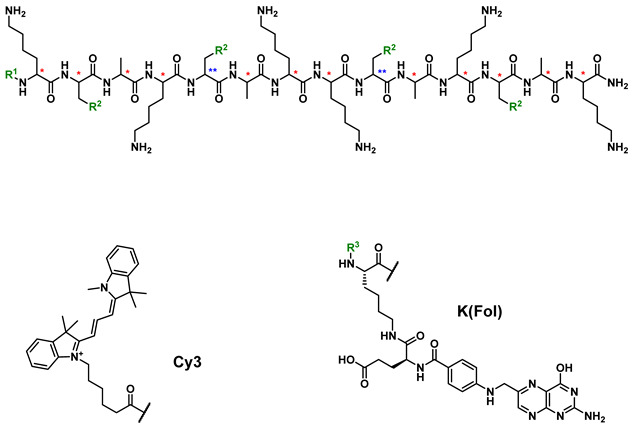									
						**EC_50_** **in µM**			
**Compound**	**R^1^**	**R^2^**	**R^3^**	*****	******	**KB**	**SK-OV-3**	**MCF7**	**HEK293**
**1**	-	-	-	-	-	36.5 (3.6)	8.6 (6.70)	110 (13.2)	221 (53.8)
**2**	H	isopropyl	-	(*S*)	(*S*)	>400	>400	>400	>400
**3**	H	isopropyl	-	(*R*)	(*R*)	331 (152)	>400	>400	>400
**4**	H	cyclohexyl	-	(*R*)	(*S*)	6.5 (0.5)	11.6 (2.37)	50.9 (3.0)	15.6 (2.6)
**5**	Cy3	isopropyl	-	(*S*)	(*S*)	6.7 (0.2)	11.0 (2.56)	7.37 (0.2)	44.3 (7.0)
**6**	Cy3	isopropyl	-	(*R*)	(*R*)	3.5 (0.1)	6.39 (0.79)	5.7 (0.7)	5.3 (0.2)
**7**	Cy3	cyclohexyl	-	(*R*)	(*S*)	5.5 (0.5)	23.0 (1.64)	11.6 (1.5)	8.2 (0.5)
**8**	K(Fol)	isopropyl	Cy3	(*S*)	(*S*)	38.8 (6.6)	8.8 (3.7)	49.4 (4.3)	35.6 (3.4)
**9**	K(Fol)	isopropyl	Cy3	(*R*)	(*R*)	22.4 (1.4)	8.0 (2.2)	13.2 (5.6)	6.9 (1.8)
**10**	K(Fol)	cyclohexyl	Cy3	(*R*)	(*S*)	3.4 (0.7)	6.0 (1.1)	6.9 (0.9)	6.8 (1.4)
**11**	K(Fol)	isopropyl	Ac	(*S*)	(*S*)	242 (33.2)	>400	>400	>400
**12**	K(Fol)	isopropyl	Ac	(*R*)	(*R*)	151 (38.7)	388.4 (21.3)	>400	>400
**13**	K(Fol)	cyclohexyl	Ac	(*R*)	(*S*)	9.8 (0.6)	41.5 (4.0)	26.9 (7.2)	20.4 (5.7)

* and ** refer to the absolute configuration of the chiral center; Ac = acetyl; pairing anion for compounds **5**–**7** and **8**–**10** is assumed to be trifluoroacetate.

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
