# Peer review of "Effect of Trimethine Cyanine Dye- and Folate-Conjugation on the In Vitro Biological Activity of Proapoptotic Peptides"

_biomolecules, 2022, doi:10.3390/biom12050725_

Round 1

Reviewer 1 Report

The manuscript entitled "Effect of trimethine cyanine dye- and folate-conjugation on the in vitro biological activity of proapoptotic peptides" written by Cardella et al. describes the synthesis and biological assay of a series of proapoptotic peptides and their conjugates with a cyanine dye and/or folate. The work is the continuation of the previous research (Nödling et al., Chem. Commun. 2020, 4672-4675). Cyanine conjugates presented in the manuscript showed increased mitochondrial targeting ability and anti-tumour activity as expected, while folate-containing constructs did not display the anticipated selectivity towards KB and SK-OV-3 cells.

The extensive synthesis of novel conjugates is well planned, there are lots of experimental results reported in this manuscript, and the overall work is interesting and could be of interest to the readers of Biomolecules.

However, some points should be addressed before publishing:

1. Although authors gave mass to charge ratios for compounds 2–13 by obtained LC-MS, they should also enclose high-resolution mass spectra (HRMS) for the aforementioned new compounds. 

2. Authors performed flow cytometry analysis to additionally analyze uptake levels of conjugates. Did they get any information about the effect of the conjugates on the cell cycle?

3. One of the major concerns of the presented study was the enhanced mitochondria targeting ability of the cyanine-peptide conjugates and the consequent anti-tumour activity. Confocal microscopy results confirmed the mitochondrial localization; however, besides confocal microscopy images, Pearson's correlation coefficients should be given for additional proof of mitochondrial localisation.

Minor points: 

  1. Although specified in the Table 1 caption, EC50 unit (μM) should also be given in the Table. 
  2. In the Chapter SPPS of compounds 5-7, part of the sentence "... or blue powder in case of labelling with compound 2" is somewhat confusing. What does it mean "labelling with compound 2" (as the labels are cyanine and/or folate)?
  3. Synthetic procedures for compound 1 and compounds 2-13 are given both in the main text and the SI, which is unnecessary duplication.

Author Response

  1. Although authors gave mass to charge ratios for compounds 2–13 by obtained LC-MS, they should also enclose high-resolution mass spectra (HRMS) for the aforementioned new compounds. 

We have analyzed compounds by high-resolution mass spectrometry. Unfortunately, our remaining sample of Ac-K(FA)-kla 12 has decomposed. This is because the first author, Davide Cardella, has finished his doctoral study at Cardiff University and is no longer in the lab, preventing us from making more conjugates for further analysis. HRMS were obtained with the remaining materials from previous confocal microscopy and flow cytometry experiments at the Shenzhen Bay Laboratory. The spectra of 2-11 and 13 are now enclosed in the Supplementary Information in Table S1.

  1. Authors performed flow cytometry analysis to additionally analyze uptake levels of conjugates. Did they get any information about the effect of the conjugates on the cell cycle?

We did not investigate the effect of conjugates on the cell cycle. We have tried to use propidium iodide to evaluate the effects of Cy3-containing conjugates on cell cycles. However, propidium iodide and Cy3 have very similar absorption and emission spectra, and our instrument has prevented us from monitoring the fluorescence of propidium iodide above 650 nm (where Cy3 fluorescence is negligible). We have also attempted to use Hoechst33258 to quantify DNA contents of cells. However, the results from Hoechst33258 and propidium iodide staining were very different in the control (i.e., non-treated) cells. In response to the editor’s request, we have decided to submit the revised manuscript without further delay.  

  1. One of the major concerns of the presented study was the enhanced mitochondria targeting ability of the cyanine-peptide conjugates and the consequent anti-tumor activity. Confocal microscopy results confirmed the mitochondrial localization; however, besides confocal microscopy images, Pearson's correlation coefficients should be given for additional proof of mitochondrial localization.

We have added Pearson's correlation coefficients in the legends of Figures 3, S4, S5, S6 and S7.

  1. Although specified in the Table 1 caption, EC50 unit (μM) should also be given in the Table. 

We have added this in Table 1.

  1. In the Chapter SPPS of compounds 5-7, part of the sentence "... or blue powder in case of labelling with compound 2" is somewhat confusing. What does it mean "labelling with compound 2" (as the labels are cyanine and/or folate)?

This has been corrected. The synthetic procedures have been removed from the manuscript and are shown only in the amended Supplementary Information. The corrected sentence now reads: “Lyophilization of the pure product fractions afforded the desired compound as red powder which were characterized by LC-MS (Figure S12-S14)”.

  1. Synthetic procedures for compound 1and compounds 2-13 are given both in the main text and the SI, which is unnecessary duplication.

The synthetic procedures have been removed from the main text and are only shown in the Supplementary Information now.

Reviewer 2 Report

Peptide conjugates constitute emerging targeted therapeutics, which combine biochemical and biological properties for future clinical applications, mainly for design of anticancer therapies. With this is mind, this manuscript described and compared the cytotoxicity of a series of proapoptotic peptides and their conjugates with a cyanine dye and/or a folate. Briefly, the findings are novel and interesting, but the manuscript needs to be revised before the publication.

  1. References must follow the journal format. Review instructions for authors.
  2. The abstract must be improved. Results should be specified more clearly with more relevant values.
  3. The authors include a lot of methodology information in the first paragraphs of the results. Authors should focus on their findings. The methodology should be described in the corresponding section.
  4. The synthesis yield, the purity of the synthetic products, the molecular mass determined by mass spectrometry could be included as a table in the study. The chromatograms presented as supplementary material evidences the high purity. However, in comparative studies it is relevant to calculate and detail the purity.
  5. Selectivity is an essential parameter in the context of clinical translation. The authors emphasize the potency. The selectivity of synthetic compounds is poorly explored. Based on this, the authors should calculate and add the selectivity index to the table describing cytotoxicity and discuss the results.
  6. How many independent experiments were performed? The IC50 deviation must be included.
  7. Statistical analysis should be included in tables and figures.
  8. A drug of reference should be included in cytoxicity assays.

Author Response

  1. References must follow the journal format. Review instructions for authors.

References have now been updated according to the journal format.

  1. The abstract must be improved. Results should be specified more clearly with more relevant values.

The abstract has now been amended. Changes are highlighted.

Despite continuous advances, anticancer therapy still faces several technical hurdles such as selectivity on cellular and subcellular targets of therapeutics. Toward addressing these limitations, we have combined the use of proapoptotic peptides, trimethine cyanine dye and folate to target the mitochondria of tumor cells. A series of proapoptotic peptides and their conjugates with a cyanine dye and/or a folate were synthesized on solid-phase, and their toxicity in different human cell lines were assessed. Cyanine-bearing conjugates were found to be up to 100-fold more cytotoxic than the parent peptides and to localize in mitochondria. However, addition of a folate motif did not enhance the potency or selectivity of the resulting conjugates toward tumor cells that overexpress folate receptor α. Furthermore, while dual-labelled constructs were also found to localize within the target organelle, they were not generally selective towards folate receptor α positive cell lines in vitro.

  1. The authors include a lot of methodology information in the first paragraphs of the results. Authors should focus on their findings. The methodology should be described in the corresponding section.

Details regarding synthesis of compounds 2-7 have been deleted, while we prefer to keep some of the discussion regarding the synthetic procedures in the section as we think it might make our rationale for the preparation of the constructs clearer to the reader.

  1. The synthesis yield, the purity of the synthetic products, the molecular mass determined by mass spectrometry could be included as a table in the study. The chromatograms presented as supplementary material evidences the high purity. However, in comparative studies it is relevant to calculate and detail the purity.

The product yields have been added to the corresponding synthetic procedures in the SI. Calculated and found m/z obtained by HRMS and purity are shown in Table S1.

  1. Selectivity is an essential parameter in the context of clinical translation. The authors emphasize the potency. The selectivity of synthetic compounds is poorly explored. Based on this, the authors should calculate and add the selectivity index to the table describing cytotoxicity and discuss the results.

A figure (Figure 2 in the amended manuscript) has been added where selectivity upon folate conjugation is reported and statistically analyzed.

  1. How many independent experiments were performed? The IC50 deviation must be included.

As stated in the Material and Methods section, three independent experiments (i.e., biological replicates) for each compound in the indicated cell line. This information has also been added to the legend of Table 1. As EC50 was not calculated for each independent experiment, yet on a set including all data from the three independent experiments, we found more appropriate to report the error on the fitting curve instead (see Table 1, values in brackets to the right of the corresponding EC50 value).

  1. Statistical analysis should be included in tables and figures.

As it would have not been practical to add statistical analysis in Table 1, statistical analysis on cytotoxicity has been included in a new figure (Figure 2) for relevant compounds. We have also amended Figure 4 to include statistical analysis.

  1. A drug of reference should be included in cytotoxicity assays.

We used Cy3 as the reference for cytotoxicity. During the revision period, we have also attempted to obtain the EC50 values of cisplatin in SK-OV-3 and MCF7. However, the reported EC50 values (ca. 10 µM, doi: 10.1186/1475-2867-10-32) of cisplatin in SK-OV-3 and MCF7 were obtained after 48-h incubation, whereas we only incubated our compounds with cells for 24 h, and cisplatin seems to have much higher EC50 value from 24-h incubation, diminishing its suitability as a reference drug. In response to the editor’s request, we have decided to submit the revised manuscript without further delay. 

Round 2

Reviewer 1 Report

All questions and points have been adequately commented on, so the manuscript can now be accepted for publication in the present form.

Reviewer 2 Report

Authors have adequately adressed most of my concerns. Based on this, I consider this version of manuscript suitable for publication.